# triCAM: A Real Monocular Multi-Modal Event-based Pedestrian Dataset

## Abstract

Event-based visions offer key advantages, such as low latency, high dynamic range, and microsecond temporal resolution. These strengths have motivated extensive research into their complementarity with other modalities, which led to the creation of several multi-modal event-based datasets. However, most of these datasets are designed for automotive or robotic domains, with limited attention to human-centered perception in everyday settings. In this paper, we introduce triCAM, a real-world monocular multi-modal event-based pedestrian dataset. triCAM integrates event streams, RGB images, depth images, IMU data, and pedestrian bounding box annotations. This dataset contains 20 sequences, each recorded in two different restaurants in both static and dynamic camera motions. By providing a rich dataset on pedestrian activities in socially interactive environments, triCAM contributes to the advancement of research in robust perception and human interaction understanding.

## 1 Introduction

Event cameras introduce a revolutionary way of capturing motion in the field of computer vision. Unlike traditional cameras, which record entire scenes at fixed intervals, event cameras operate asynchronously by detecting per-pixel brightness changes. This bio-inspired sensing mechanism allows them to achieve high temporal resolution, high dynamic range, and low power consumption. These advantages have led to their adoption in domains requiring high-speed and robust perception, including robotics, autonomous vehicles, and surveillance systems. While event cameras excel at capturing fast motion and high dynamic range scenes, they inherently provide sparse information focused on intensity changes rather than full-textural scene appearance. To alleviate this limitation and facilitate support for a wider range of computer vision tasks, it is beneficial to combine event data with supplementary modalities. Accordingly, several event-based multi-modal datasets have been proposed, often combining events with auxiliary modalities such as RGB images, depth, LIDAR, calibration information, and inertial measurement units (IMU). For instance, MVSEC Zhu et al. (2018), DSEC Gehrig et al. (2021b), M3ED Chaney et al. (2023), CoSEC Peng et al. (2024), SLEDBrebion et al. (2023), and ECMD Chen et al. (2023) are popular benchmarks in automotive and robotics environments with stereo event streams and RGB images. Although these stereo multi-modal datasets contribute greatly to the vision community, their reliance on stereo multi-modal configurations introduces extra hardware cost, multi-sensor calibration complexity, and power consumption. Some researchers try to simplify these stereo datasets by using only one camera from the pair to simulate a monocular setup. But this does not adequately capture the design requirements of a true monocular multi-modal dataset.

As a result, they impose a significant computational cost which makes them unsuitable for resource-constrained applications and provides limited insight into socially interactive environments. In contrast, common places like restaurants remain unexplored, even though understanding pedestrian interactions and motion patterns in such cluttered, dynamic settings is essential. To address this gap, we propose triCAM, a real, monocular, multi-modal dataset targeting pedestrians in restaurant environments. triCAM integrates data from multiple modalities, event streams, RGB images, and depth images (see Figure 1) in addition to the IMU data, pedestrian bounding box annotations, and the sensors' calibration parameters. This dataset was recorded by three sensors, an event camera, a RGB-D camera and an IMU sensor, as displayed in Figure (a) of Table 1.

In summary, triCAM offers several key contributions to the field of event-based multi-modal vision:

1. It is the first publicly available multi-modal monocular dataset designed for pedestrian-centered scenarios in both outdoor and indoor restaurant environments.

2. It represents a multi-modal event-based dataset comprising complementary modalities RGB, depth, IMU data, calibration parameters, and pedestrian bounding boxes.

3. It provides spatially and temporally aligned sequences recorded under both static and dynamic camera motions.

By focusing on enclosed social environments, triCAM uniquely complements existing datasets. It opens new directions in event-based vision research, particularly for applications involving human-centered perception and socially interactive contexts such as human behavior analysis, service robot navigation, occupancy detection, and human-robot interaction (HRI) for delivery robots. Furthermore, triCAM can be utilized to perform a variety of tasks, including monocular depth estimation, pedestrian detection, ego-motion estimation, and multi-modal model training.

Table 1: triCAM hardware descriptions. Figure **(a)** The camera setup showing the triCAM sensor rig arrangement and Table **(b)** its hardware specifications with detailed information about the sensors, their parameters, and key characteristics.

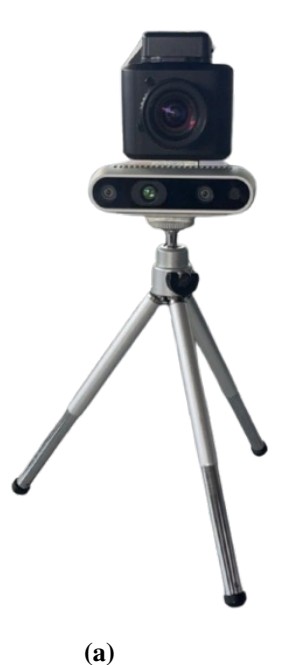

| Sensors | Description |
|---|---|
| **1X WitMotion IMU** | 200 Hz
3-axis Accelerometer
3-axis Gyroscope
3-axis Magnetometer
4-axis Quaternion
Roll, Pitch, Yaw |
| **1X Prophesee Gen3** | Resolution: $640 \times 480$
3/4" CMOS
Monochrome
$\geq$120 dB dynamic range |
| **1X RealSense D435i** | **RGB:** monoscopic
Resolution: $1920 \times 1080$
FOV: 69°H / 42°V
frame rate: 30 fps
**Depth:** stereoscopic
FOV: 87°H / 58°V
Resolution: $1280 \times 720$
frame rate: up to 90 fps
**IMU:** 63 Hz & 200 Hz
3-axis Accelerometer
3-axis Gyroscope |

|  (a)  |  (b)  |
|---|---|

## 2    RELATED WORK

Table 2 summarizes the characteristics of both stereo and monocular multi-modal event-based datasets. In this section, we review and compare these existing datasets in detail.

### 2.1    STEREO DATASETS

A popular dataset in event-based vision is MVSEC Zhu et al. (2018). MVSEC was the first large-scale stereo dataset for event-based vision. Its sensor rig is composed of a pair of DAVIS m346B event cameras ($346 \times 260$) with a baseline of 10 cm, a VI-Sensor with a stereo RGB camera, capturing both indoor and outdoor driving and drone scenarios. It records data from a variety of vehicles, including cars, motorbikes, hexacopters, and handheld devices.

Table 2: Comparison of existing multi-modal event-based datasets.

| Datasets | Type | Events | RGB | Depth | IMU | Env | Scenarios |
|---|---|---|---|---|---|---|---|
| **Stereo** | | | | | | | |
| MVSEC Zhu et al. (2018) | Real | ✓ | ✓ | ✓ | ✓ | Both | Automotive |
| DSEC Gehrig et al. (2021b) | Real | ✓ | ✓ | ✓ | ✓ | Outdoor | Automotive |
| VECTOR Gao et al. (2022) | Real | ✓ | ✓ | ✓ | ✓ | Indoor | Diverse |
| M3ED Chaney et al. (2023) | Real | ✓ | ✓ | ✓ | ✓ | Both | Robotics |
| CEAR Zhu et al. (2024) | Real | ✓ | ✓ | ✓ | ✓ | Both | Robotics |
| **Monocular** | | | | | | | |
| D-eDVS Weikersdorfer et al. (2014) | Real | ✓ | ✓ | ✓ | | Indoor | Robotics |
| DDD17 Binas et al. (2017) | Real | ✓ | ✓ | | | Outdoor | Automotive |
| VINS-Mono Qin et al. (2018) | Real | ✓ | ✓ | | ✓ | Both | Robotics |
| CED Scheerlinck et al. (2019) | Real | ✓ | ✓ | | | Both | Automotive |
| EventCap Xu et al. (2020) | Real | ✓ | ✓ | ✓ | | Indoor | Robotics |
| DENSE Hidalgo-Carrio et al. (2020) | Synthetic | ✓ | ✓ | ✓ | | Outdoor | Automotive |
| EventScape Gehrig et al. (2021a) | Synthetic | ✓ | ✓ | ✓ | | Outdoor | Automotive |
| Agri-EBV Zujevs et al. (2021) | Real | ✓ | ✓ | ✓ | ✓ | Outdoor | Agriculture |
| TUM-VIE Klenk et al. (2021) | Real | ✓ | ✓ | | ✓ | Indoor | Robotics |
| MonoANC Shi et al. (2023) | Synthetic | ✓ | ✓ | ✓ | | Indoor | Automotive |
| RGB-Event ISP Yunfan et al. (2024) | Real | ✓ | ✓ | | | Outdoor | ISP |
| HUE Ercan et al. (2024) | Real | ✓ | ✓ | | | Both | Automotive |
| **triCAM (ours)** | Real | ✓ | ✓ | ✓ | ✓ | Both | Restaurant |

It fuses this data with LiDAR, a nine-axis IMU, motion capture, and GPS to provide ground-truth pose and depth images. MVSEC is a key dataset for depth and odometry benchmarking. Another one is DSEC Gehrig et al. (2021b), which expanded scale in the automotive sector by providing high-resolution stereo events with a pair of Prophesee Gen3.1 cameras ($640 \times 480$) with a baseline of 60 cm with two RGB cameras in outdoor driving scenarios in the city of Zurich.

With additional 16-channel LiDAR and IMU data, DSEC is a widely used benchmark for event-based stereo depth estimation due to its precise calibration and large-scale sequences. VECTOR Gao et al. (2022) shifts attention from driving to indoor robotics, integrating Prophesee Gen3 stereo cameras ($640 \times 480$), stereo RGB cameras ($1224 \times 1024$), a LiDAR, and a nine-axis IMU. Collected in structured indoor environments, it supports SLAM and localization under controlled but dynamic human-centric conditions, broadening event-based applications beyond automotive use cases. M3ED Chaney et al. (2023) targets robotics applications by recording data from both forest and urban environments using ground, aerial, and legged robots. Alongside stereo event cameras ($1280 \times 720$) and RGB cameras ($1280 \times 800$), the dataset includes LiDAR and IMU, supporting perception tasks in unstructured and dynamic scenarios. ME3D is suited for robotic navigation and mapping tasks. Finally, CEAR Zhu et al. (2024) pushed stereo event datasets further with a strong focus on agile quadruped robots. With stereo event cameras combining DAVIS 346 and DVXplorer Lite, RGB-D, LiDAR, and a 12-axis IMU sensor, CEAR captures indoor and outdoor sequences under rapid motion where conventional cameras fail due to blur, making it the first dataset focused explicitly on agile event-based robotics.

## 2.2 MONOCULAR DATASETS

### 2.2.1 SYNTHETIC DATASETS

EventScape Gehrig et al. (2021a) is a simulated multi-modal dataset. This dataset provides large-scale asynchronous event streams generated from the CARLA simulator Dosovitskiy et al. (2017), rendered at 500 Hz, and converted into events via an event simulator tool, ESIM Rebecq et al. (2018). Each event arises from pixel-wise brightness changes simulated from the rendered RGB images, and it also includes depth images, semantic segmentation and vehicle navigation parameters, making it an ideal benchmark for automotive scenarios. Its focus on tightly synchronized RGB and event data establishes a foundation for multi-modal perception research. Building upon this idea of simulated multi-modal data, the DENSE dataset Hidalgo-Carrió et al. (2020) further explores event-RGB integration.

Like EventScape, it uses CARLA Dosovitskiy et al. (2017) for data generation, but the virtual event camera is modeled after the DAVIS346B sensor with a resolution of $346 \times 260$ pixels. Recorded at 30 frames per second, DENSE provides depth images, RGB images, and simulated event streams under diverse lighting and weather conditions. This allows researchers to study event-driven perception in controlled yet varied environments.

Lastly, MonoANC Shi et al. (2023) extends these efforts into more challenging driving conditions. Specifically designed to tackle night-time scenarios and adverse weather, MonoANC offers 11,191 samples of synchronized RGB, event, and depth data. Its multi-modal nature also supports research into robust event-RGB integration. By emphasizing asynchronous events combined with frame-based data, MonoANC demonstrates the value of multi-modal approaches for perception in low-light and dynamic conditions.

### 2.2.2 REAL DATASETS

The first event-based multi-modal dataset is D-eDVS Weikersdorfer et al. (2014). This dataset's sensor rig is composed of a PrimeSense RGB-D camera and an e-DVS. The eDVS operated at a resolution of approximately $128 \times 128$ eDVS event camera to capture asynchronous events, while the PrimeSense sensor provided synchronized RGB and depth data by atrgetting only robotics applications. The DDD17 Binas et al. (2017) dataset captured automotive data with a DAVIS346B sensor, which outputs both events and active pixel sensor (APS) frames at $346 \times 260$ pixels. No depth sensor or IMU data were provided, but the dataset included vehicle telemetry such as steering angle, throttle, brake, and GPS. It was designed for outdoor automotive perception in challenging driving conditions. Another widely used dataset is VINS-Mono Qin et al. (2018), for monocular visual-inertial odometry (VIO). It employed a rolling-shutter monocular camera with a resolution of $752 \times 480$ pixels, complemented by a 9-axis IMU providing accelerometer, gyroscope, and magnetometer data. The dataset spans both indoor and outdoor robotics environments. Scheerlinck et al. (2019) presented a colored event cameras dataset (CED). This dataset was collected using a color-DAVIS346 sensor (resolution $346 \times 260$), which provides both real events coupled with synthetic colored events generated by ESIM Rebecq et al. (2018) and ground-truth RGB images. CED primarily focused on automotive and robotics navigation in indoor settings. EventCap Xu et al. (2020) introduced a revolutionary way of capturing 3D human motion using a DAVIS240C event camera ($240 \times 180$) along with their generated intensity frame from the same camera. This dataset provides object-wise depth images for human pose estimation. The human actions were recorded with a Sony RX0 camera, which produces high frame rate (between 250 and 1000 fps) RGB videos at $1920 \times 1080$ resolution. This dataset consists of 12 sequences of 6 actors performing different activities, including karate, dancing, javelin throwing, and boxing. The dataset covers indoor robotics scenarios, with an emphasis on human motion and interaction. HUE Ercan et al. (2024) is a high-resolution multi-modal dataset collected with a Prophesee Gen4M with a resolution of $1280 \times 720$ and Allied Vision Alvium compact CMOS cameras with a resolution of $1456 \times 1088$. This dataset contains only RGB images and event streams and was primarily designed for indoor automotive and robotics applications under low-light and high-dynamic-range conditions. The RGB-Event ISP dataset Zujevs et al. (2021) provided pixel-aligned RAW images and event streams captured with a hybrid vision sensor from a monocular viewpoint. It contains over three thousand samples across diverse scenes, lighting conditions, exposures, and lenses, with color calibration generated by a ColorChecker. Unlike previous event datasets that mainly target high-level vision tasks, this dataset is designed to support research on event-guided image signal processing (ISP). Zujevs et al. (2021) presented their work titled *"An Event-based Vision Dataset for Visual Navigation Tasks in Agricultural Environments"*. Agri-EBV is a dataset designed for agricultural robotics featuring different agricultural environments. It used a DAVIS240 camera ($240 \times 180$), a RealSense RGB-D depth camera, LIDAR-16, and an IMU for inertial measurements. This dataset uniquely emphasizes outdoor crop monitoring and agricultural tasks under challenging movement in a rural area. While multi-modal event-based datasets reviewed above provide an important contribution, they largely overlook pedestrian-centered scenarios in social and crowded environments. Although Pedro Boretti et al. (2023) is a monocular event-based pedestrian dataset, it lacks other modalities to expand research in this area.

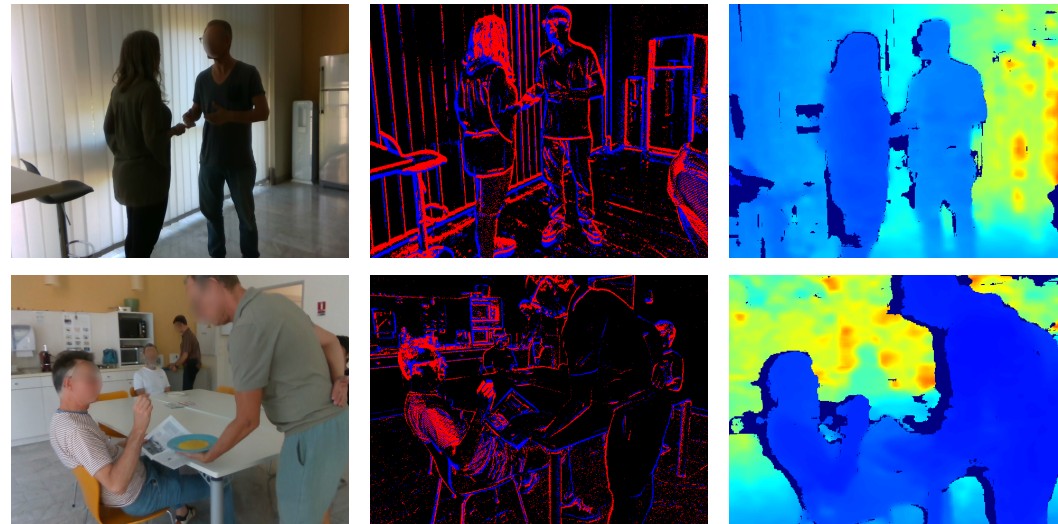

Figure 1: Overview of triCAM sequences with RGB, event, and depth modalities.

## 3 HARDWARE SETUP

The triCAM sensor rig consists of a dual camera setup with an additional an IMU sensor, as displayed in Figure (a) of Table 1. The depth, RGB images, and IMU data were captured by a RGB-D RealSense D435i depth camera, the event streams from a Prophesee Gen3 camera, and the additional IMU data from a WitMotion sensor.Table 1 contains detailed information about the characteristics of each sensor. These sensors are mounted on a standard tripod for both static and dynamic camera motions.

In the following section, we describe the data post-processing pipeline, synchronization across cameras, and calibration parameters extraction.

### 3.1 HARDWARE AND SOFTWARE

The triCAM data acquisition pipeline was managed through a custom graphical user interface (GUI) application developed using Tkinter Lundh (1999). Therefore, the cameras were connected to a laptop during the recording period. The GUI application facilitated user interaction and controlled a backend program responsible for coordinating the sensors. Specifically, the backend handled scene metadata storage, triggered simultaneous recording of the two cameras and the IMU via a multi-threaded process. Once the recordings were captured, several post-processing steps were performed entirely in Python. First, all the images were resized to the same resolution of 640x480 pixels. The RealSense SDK 2.0 Schmidt et al. (2019) was employed to temporally and spatially align the depth and RGB images by accounting for their distinct fields of view and to extract timestamps for each of these modalities. Camera calibration was carried out using the OpenCV Calib library, while pedestrian detection and annotation of RGB images were semi-automatically generated using YOLOv8x Hussain (2023).

### 3.2 TIME SYNCHRONIZATION

The synchronization of the RGB and depth images from the RealSense camera was straightforward, as these images were temporally aligned and spatially preprocessed using the RealSense SDK 2.0 Schmidt et al. (2019). However, temporally aligning the event streams with the RGB and depth frames required more processing due to the event camera's continuous and asynchronous output. Therefore, to achieve a temporal correspondence between the events and the other modalities, the start time of the event camera recording and the timestamps of each depth frame were recorded. Each event timestamp was converted to a global reference by adding the event camera's start time.

Since the RGB-D camera operated at approximately 30 FPS (one frame every 33 ms), the event streams were segmented into 33 ms intervals corresponding to consecutive depth timestamps on a global timeline to align the two modalities temporally. For each depth frame, all events that occurred between its timestamp and the next were aggregated. In this way, each depth frame was synchronized with its corresponding events in time. To achieve temporal synchronization between the two IMUs, RGB-D, and event cameras, the depth frame timestamps from the RealSense RGB-D camera were used again as the reference timeline. This is because the IMU measurements from the Realsense D435i camera are timestamped using the depth sensor's hardware clock, ensuring that accelerometer and gyroscope readings are already aligned with the depth frames. Since the IMU of this dataset had different sampling rates with 200 Hz for the event camera IMU, 63 Hz for the RGB-D accelerometer, and 200 Hz for the RGB-D gyroscope as showcased in Table 1. All signals were resampled using linear interpolation to a high frequency of 1 kHz timeline covering the duration of the recording. Any systematic temporal offsets between the two IMU and depth frames were then estimated using cross-correlation of motion signals, and the IMU timestamps were adjusted accordingly. Finally, to keep everything in sync, we grouped the IMU samples that fell within each frame interval for every modality, aligning the IMU, RGB-D, and event streams on a shared timeline.

### 3.3 Spatial Synchronization

The multi-modal content of this dataset was spatially synchronized using the OpenCV Calib library to extract both intrinsic and extrinsic calibration parameters of each camera. These calibration parameters were generated from a 12x8 checkerboard grid with a square size of 30 mm. This calibration pattern was captured in various rotations and positions to ensure robust calibration results. For the RGB camera, calibration was performed directly on grayscale images of the checkerboard grid. While for the event camera, we followed the calibration pipeline proposed by Muglikar et al. (2021). Following this approach, the event streams were first transformed into image-like representations by aggregating events over brief temporal windows of 33.33 ms to ensure temporal synchronization with the RGB-D camera and then reconstructing them into grayscale event frames using a pretrained event-to-video E2VID model (Rebecq et al., 2019a;b). The resulting grayscale event frames were then used together with the RGB frames to estimate both intrinsic camera parameters and extrinsic parameters with the projection of the event camera to the RGB-D camera. Finally, an additional pixel-wise warping algorithm was applied to achieve precise spatial alignment between the events and RGB images.

As a result, triCAM consists of two types of event data, the raw event streams and the rectified, spatially aligned event streams paired with the RGB images. Figure (c) 2 illustrates the overlay of rectified events on RGB images.

## 4 Dataset

### 4.1 Dataset Labeling

The triCAM dataset was collected using two distinctive cameras, a RGB-D and an event camera and contains RGB images, depth images and both raw and rectified events. To encourage multi-modal learning as well as mono-modal learning, this dataset consists of two bounding box annotations for each modality namely the RGB and raw event data. Moreover, the rectified events share the same bounding boxes as the RGB images as displayed in Figure (c)2. Given that the RGB and depth data are spatially and temporally aligned, the RGB bounding boxes correspond perfectly to the depth ones. The RGB image labeling process was done semi-automatically, the images were first annotated automatically using the pretrained object detection model YOLOv8x Hussain (2023). However, the results were unreliable due to the low resolution of the images and the clustered nature of the pedestrians in the scene, as displayed in Figure (a) 2. Therefore, the first results were double-checked manually to ensure high-quality annotations using one of the most popular image annotation tools, LabelImg Tzutalin (2015). On the other hand, the event streams were converted into an image-like representation to generate their corresponding pedestrian bounding boxes, as showcased in Figure (b) 2. This process was performed entirely manually because the labeling tool failed to detect pedestrians in the event frames, owing to their non-textural nature particularly in static scenarios where the events does not sufficiently reveal the scene content. Figure 2 demonstrates some annotations results on both static and dynamic sequences of the same scene.

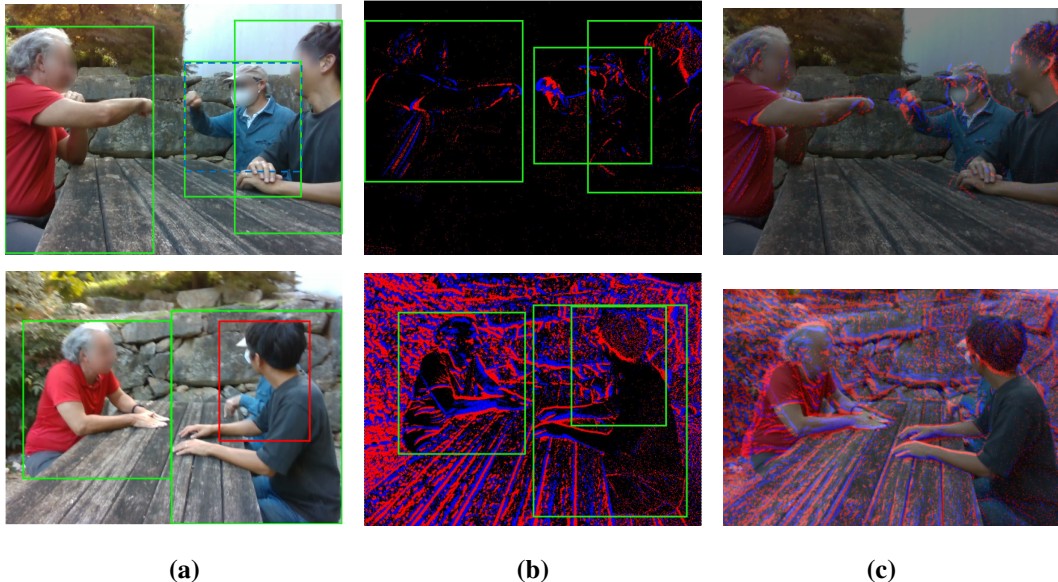

(a)                                    (b)                                    (c)

Figure 2: triCAM data labeling results for one sequence, shown for both a static scene (top row) and a dynamic scene (bottom row). Each column are categorized into three images:**Figure (a)** RGB annotations, where green bounding boxes denote the generated ground truth, red boxes indicate missed annotations, and blue dashed boxes represent YOLOv8x predictions,**Figure (b)** manually annotated pedestrian bounding boxes on the event data, and **Figure (c)** an overlay of the rectified event stream on the corresponding RGB image.

## 4.2 DATASET FORMAT

triCAM is distributed in a zip file format to ensure easy accessibility to large research communities. Each sequence contains synchronized modalities, including RGB images, depth images, event streams, each camera's IMU data, and the calibration parameters and pedestrian bounding boxes. The bounding boxes are provided for both image and event streams in YOLO format. The RGB and depth images are saved in PNG format, while the raw and rectified event streams are stored as NumPy arrays. Additionally the camera calibration parameters are provided in YAML format, alongside the IMU data saved as CSV files.

## 4.3 DATASET SEQUENCES

The triCAM dataset has 20 sequences captured in two distinct restaurants with people going about their usual activities, such as eating, drinking, chatting, walking between tables and interacting with waiters, as illustrated in Figure 1. The special motion and interaction patterns displayed by each activity capture the organic dynamics of a busy outdoor and indoor with low-light restaurant settings. The dataset features a group of participants aged from 20 to 70 of diverse ethnicities to provide a rich variety of manners and behaviors. Table 3 summarizes how each activity was documented as a distinct sequence for clarity because each sequence name is composed of the restaurant, the major pedestrians' activities, the environment and the occlusion level. The dataset includes recordings captured under both static and dynamic camera motions. For static scenarios, the camera is fixed at a single location while for dynamic camera motions, the camera is handheld allowing fast motion across the scene. As expected, the rapid movement in the dynamic sequences produced significantly more events than the static ones, as illustrated in Figure 2.

All participants involved in the data collection process provided informed consent prior to participation. They were fully aware that their data would be recorded and included in a publicly available dataset for research purposes. The data were collected in accordance with the ethical protocol of our institution's GDPR.

To further protect their privacy, we applied a face detector YOLOv12L Hussain (2023) pretrained model on all RGB images and applied a Gaussian mask on all detected faces to blur each participant's face. This process was thoroughly analyzed to ensure the privacy of our participants. Currently, we refrain from sharing the dataset website due to the double-blinding review procedure of this conference.

Table 3: The triCAM sequences details of one of the restaurants. Each sequence was recorded in two camera motions. *Static* with the camera fixed on a table and *Dynamic* with the camera handheld in constant motion. **Occlusion** is the occlusion level of the persons in the scene,**Time** represents the duration, **Persons** indicates the number of people, and **Events** shows the total number of events generated in each sequence.

| Sequences | Camera Motion | Occlusion Level | Time (s) | Persons | Events (M) |
|---|---|---|---|---|---|
| R1_walk_in_01 | Static | high | 172 | 8 | 54 |
| | Dynamic | high | 180 | 8 | 140 |
| R1_walk_in_02 | Static | low | 182 | 4 | 234 |
| | Dynamic | low | 130 | 4 | 400 |
| R1_sit_eat_out_01 | Static | high | 185 | 10 | 302 |
| | Dynamic | high | 190 | 10 | 385 |
| R1_sit_eat_out_02 | Static | high | 185 | 8 | 72 |
| | Dynamic | high | 190 | 8 | 105 |
| R1_sit_eat_in_01 | Static | medium | 205 | 6 | 340 |
| | Dynamic | medium | 160 | 6 | 545 |
| R1_interact_in_01 | Static | high | 195 | 7 | 100 |
| | Dynamic | high | 200 | 7 | 195 |
| R1_interact_out_01 | Static | low | 195 | 4 | 198 |
| | Dynamic | low | 200 | 4 | 790 |
| R2_carry_out_01 | Static | low | 178 | 5 | 230 |
| | Dynamic | low | 185 | 5 | 418 |
| R2_carry_out_02 | Static | low | 178 | 3 | 120 |
| | Dynamic | low | 185 | 3 | 232 |
| R2_chat_in_01 | Static | medium | 178 | 6 | 53 |
| | Dynamic | medium | 185 | 6 | 187 |

## 5 EXPERIMENT RESULTS

We conducted two experiments on both static and dynamic subsets of the triCAM dataset. The first experiment focuses on an in-depth pedestrian detection analysis, and the second one on a monocular depth estimation to assess the performance of pixel-wise distance prediction.

### 5.1 PEDESTRIAN DETECTION

Table 4: Baseline results for pedestrian detection using YOLOv8x.

| Motion | Modality | $mAP_{50}$ | $mAP_{50:95}$ |
|---|---|---|---|
| Static | Event-only | 0.275 | 0.136 |
| | RGB-only | 0.543 | 0.381 |
| | RGB+Event | 0.626 | 0.404 |
| Dynamic | Event-only | 0.390 | 0.252 |
| | RGB-only | 0.427 | 0.315 |
| | RGB+Event | 0.659 | 0.422 |

The pedestrian detection on the triCAM dataset sequences was evaluated using the pretrained YOLOv8x modelHussain (2023).We trained the dataset on the R1 restaurant sequences and tested on the R2 sequences. Each motion group was evaluated across three modalities, Event-only and RGB-only and hybrid data with RGB+Event. The latter is a result of a prediction combination of both event-only and RGB-only models using a late fusion approach.

Where each model produced its own set of bounding boxes, which were then merged using Non-Maximum Suppression (NMS) Bodla et al. (2017). The event streams were converted into voxel-grids of 3 channels for convenience with the pretrained model input expectation. Each motion-based dataset was partitioned into 5 training, 2 validation, and 3 testing sequences. The model training was performed with images resized to 640×480 with a batch size of 16 for 50 epochs.

To evaluate the pretrained model across all sequences, we use mean Average Precision (mAP) Lin et al. (2015), reported using two metrics, $mAP_{50}$ and $mAP_{50:95}$. As shown in Table 4, in the static setting, RGB-only already provides a strong baseline, but combining RGB and events significantly boosts performance. This result is largely because the camera is fixed, resulting is sharp RGB images with few events, since events are produced only when motion occurs in the scene. While under dynamic handheld motion, both modalities experience performance degradation, which is expected due to motion blur in RGB frames and the more complex event patterns generated during rapid movement. Event-only performance improve considerably, while RGB-only degrades. But their fusion remains the most robust across all conditions.

## 5.2 MONOCULAR DEPTH ESTIMATION

Table 5: Baseline results for monocular depth estimation using HMnet.

| Motion | Modality | Errors ↓ | | | Accuracy ↑ | | |
|---|---|---|---|---|---|---|---|
| | | Abs.Rel | RMSE | RMSElog | $\delta < 1.25$ | $\delta < 1.25^2$ | $\delta < 1.25^3$ |
| Static | Event-only | 0.524 | 7.91 | 0.485 | 0.402 | 0.563 | 0.675 |
| | RGB+Event | 0.318 | 5.42 | 0.292 | 0.642 | 0.801 | 0.911 |
| Dynamic | Event-only | 0.267 | 2.74 | 0.152 | 0.748 | 0.842 | 0.941 |
| | RGB+Event | 0.196 | 2.11 | 0.114 | 0.823 | 0.912 | 0.966 |

Table 5 presents the baseline results of monocular depth estimation using the pretrained HMnet B3 model Hamaguchi et al. (2023). This table indicates the impact of motion in depth estimation through a clear distinction between static and dynamic motion performance results. Under static settings, due to the minimal of event activity, the Event-only performance is limited. However, RGB+Event outperforms Event-only with higher accuracy and lower depth errors by combining sparse event motion cues with rich RGB information. In contrast, dynamic scene boost events performance across all metrics but RGB+Events achieve the best overall performance. This result demonstrates that the combination of motion cues from events and textual information from RGB provide complementary benefits by reducing depth errors. The consistent motion in the scene allows the network to extract reliable depth cues from events and RGB images. Overall, these results highlight how fast handheld motion introduces significant information for monocular event-based and hybrid depth estimation.

From these results, we conclude that the triCAM dataset provides a versatile benchmark for both pedestrian detection and monocular depth estimation. Its variety in camera motion allows systematic benchmarking across different tasks. This diversity also encourages the design of models that can adapt to different levels of scene dynamics and supports research into multi-modal fusion that leverage the complementarity of RGB and events data.

## 6 CONCLUSION

In this paper, we introduce triCAM, the first monocular, multi-modal, event-based pedestrian dataset. Designed for real-world applications, triCAM provides high-quality, synchronized data collected in both low-light indoor and outdoor restaurant environments, under static and dynamic camera motions. Unlike existing datasets, it captures natural human interactions in crowded scenes, offering a unique benchmark for studying pedestrian detection and human behavior. By combining complementary sensing modalities, triCAM enables robust representation learning and open new opportunity for advancing in event-based perception.

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
