# OpenReview forum: "triCAM: A Real Monocular Multi-Modal Event-based Pedestrian Dataset"
_ICLR.cc/2026/Conference — Submitted to ICLR 2026_

### Official Review · Reviewer_9EHV · 2025-10-27

**Soundness:** 2
**Presentation:** 2
**Contribution:** 2
**Rating:** 2
**Confidence:** 4

**Summary:**

This paper introduces triCAM, an event-based, real-world monocular multimodal pedestrian dataset. The dataset integrates event streams, RGB images, depth images, IMU data, and pedestrian bounding box annotations recorded in indoor and outdoor restaurant environments under both static and dynamic camera motions.

**Strengths:**

This type of dataset seems uncommon in the event camera field.

**Weaknesses:**

1. Does PEDESTRIAN only cover people in restaurants? According to the KITTI rankings, PEDESTRIAN should also cover people in open environments like roads.

2. Is this dataset's task just pedestrian detection? I'd prefer to see results on depth estimation, event camera SLAM, or motion prediction. This results in the experimental section being too sparse, with tables and tables spanning less than a page. I'd prefer to see a more comprehensive, multi-task benchmark.

3. The TRICAM dataset is too small. The 20 sequences total only 60 minutes of data, which is far from sufficient for a portable dataset that doesn't require any motion capture. Furthermore, it was captured in just two restaurants, so the environment and lighting diversity are limited compared to large-scale benchmarks.

3. I have serious questions about the calibration results for the RGB and event cameras. In particular, I don't think the reconstructed checkerboard is good in Figure 3. I hope the authors can provide a visualization of the overlap between the event and RGB synchronized frames.

4. I'm very surprised that this dataset collects such sensitive personal data without any privacy protection. This clearly violates ethical review rules.

**Questions:**

See Weaknesses.

---

> ### Author Response · Authors · 2025-11-26
>
> Thank you for your insightful and constructive feedback. The revised version contains your constructive suggestions. We will address the highlighted weaknesses below:
>
> 1- Yes, the triCAM dataset is a small pedestrian-centered multi-modal dataset. This dataset aims to provide valuable contributions to events and general computer vision communities. Compared to large-scale datasets such as KITTI, this dataset aims for smaller-scale applications. The goal of our dataset is not to provide a large-scale general benchmark, but a focused dataset tailored to the study of both low-light indoor and outdoor human interaction.
>
> 2- Thanks for your valuable suggestion, we conducted additional experiments on monocular depth estimation and pedestrian detection. For each task, we evaluated their performance on both static and dynamic (fast motion) subsets of every sequence to reveal the diversity and challenges of this dataset. The results are now presented in section 5, specifically 5.2, where we compare the performance across static and dynamic subsets and highlight the effect of the fast motion of the baseline result.
>
> 3- Yes, we acknowledged the small size of this dataset. For the purpose mentioned in response 1, the collected 20 sequences contain dense, high-quality, temporally rich event data, which is more important than duration, which is approximately 60 minutes. Because event cameras produce orders of magnitude more temporal measurements than conventional RGB sensors. Therefore, a one-minute recording of an event camera contains millions of asynchronous events, making it larger than the duration suggested. Further information about the number of events generated for each sequence can be found in Table 3.
>
> 4-  Regarding the overlap between the RGB and event modalities, we clarify the following points.
> a- Temporal synchronization is accurate.
> Both sensors are synchronized using timestamp alignment, ensuring that the depth and RGB frames and the corresponding event slice represent the same moment in time.
>
> b- Spatial alignment was performed using our calibrated extrinsic.
> After calibration, we apply a pixel-wise warping function using the estimated rotation and translation between the cameras. As event cameras have unique imaging characteristics, most automatic calibration pipelines struggle to extract perfectly accurate extrinsic for RGB and event pairs. Small calibration errors are common and amplify when visualized on a checkerboard, which explains the imperfections observed in Figure 3, now replaced by Figure 2 with RGB-Event overlapping images in the updated version of the article.
>
> 5- Thank you for mentioning this critical issue. We apologize for not describing the privacy-protection measurement in the previous version. Before releasing the dataset, we automatically anonymize all participants' identifiable facial features. We apply a pretrained facial detector “YoloFaceV12L” model to every RGB image and then use a Gaussian blurring mask over the detected facial bounding box. This anonymization process was double-checked to ensure the privacy of our participants is well protected. All the above-mentioned issues have been addressed in the revised version of the article from line 375 to 382.
>
> Thank you for your thoughtful feedback.

---

### Official Review · Reviewer_79fb · 2025-10-29

**Soundness:** 3
**Presentation:** 2
**Contribution:** 2
**Rating:** 4
**Confidence:** 1

**Summary:**

The paper introduces triCAM, a real-world monocular multi-modal dataset targeting pedestrian scenes in restaurant environments. triCAM contains synchronized streams from a Prophesee Gen3 event camera, an Intel RealSense D435i (RGB + depth + IMU), and an additional WitMotion IMU. The dataset comprises ~20 sequences recorded in two restaurants under both static and handheld (dynamic) camera motions, and includes calibration parameters and bounding-box annotations for both RGB and event-derived image representations. The authors describe the hardware/software setup, spatial/temporal synchronization pipeline (including E2VID reconstruction of events for calibration/annotation), dataset format (ROS bag + supplementary files), and present a baseline pedestrian detection evaluation using YOLOv8x on RGB-only, Event-only, and late-fusion Event+RGB setups. The manuscript positions triCAM as filling a gap in human-centric, socially interactive event datasets.

**Strengths:**

**Originality**: Focus on indoor/outdoor restaurant pedestrian scenes (socially interactive, cluttered) is novel among event multi-modal datasets — most prior work targets automotive or robotics navigation.

**Quality**: Use of a modern Gen3 event camera plus RealSense D435i and separate IMU provides complementary modalities; distribution in ROS bag format and inclusion of intrinsic/extrinsic calibration parameters increases usability.

**Clarity**: Paper is structured clearly with useful tables/figures (sensor specs, sequence statistics, baseline results) that help readers understand dataset composition.

**Significance**: Enables research directions (multi-modal pedestrian detection, event-guided pose/depth estimation, perception in socially interactive settings) that are underexplored in event vision.

**Weaknesses:**

1. Figure/table layout and information density are insufficient. The paper’s sequence overviews and calibration/annotation examples (Figures 2, 3, 4) present a formal view but lack more details, so readers cannot reliably assess annotation or reconstruction quality.
2. Dataset statistics and sample-level examples are incomplete. Although Table 3 lists per-sequence duration, number of people, and total events, the manuscript misses key sample-level statistics (e.g., per-frame or per-sequence bounding-box count distributions, frame counts per distance bin, quantitative occlusion tiers) and lacks representative “difficult sample” parallel visualizations that would let readers judge diversity and difficulty.
3. Lacks clear problem definition and theoretical explanation. While the introduction and contribution sections state triCAM’s purpose and novelty, the manuscript does not formalize the tasks or evaluation objectives (for example, concrete definitions of the “multimodal fusion” problem being solved and the evaluation criteria).
4. Experimental reporting is insufficient (narrow baselines, missing ablations and per-condition breakdowns). The evaluation uses YOLOv8x variants (Event-only, RGB-only) and a simple late-fusion, reporting only overall mAP50/Precision/Recall (Table 4). There are no per-condition results (static vs. dynamic, occlusion levels, distance bands), nor ablations on key hyperparameters (e.g., the 33.33 ms event aggregation window).

**Questions:**

1. Improve the layout of Figures 2–5 to make the information more readable (e.g., clearer captions)
2. Consider presenting diversity metrics, for example: variation in lighting, motion, and occlusion; number of unique subjects; and environmental diversity (indoor/outdoor).
3. Formally define the tasks that triCAM aims to support (e.g., “multimodal monocular detection,” “event–RGB fusion tracking,” etc.)
4. Add ablation studies on key parameters such as the event aggregation window (33.33 ms).
5. Provide per-condition analyses, such as static vs. dynamic scenes, different occlusion levels, lighting conditions, and distance ranges.

---

> ### Author Response · Authors · 2025-11-27
>
> Thank you for your detailed feedback. We will address the highlighted weaknesses below:
>
> 1- We have improved all the figures with clearer captions and additional annotations to make sequence overviews, calibration, and annotation examples more readable. This issue has been addressed and kindly check Figure 2 in the updated version to observe the new changes.
>
> 2- Due to page limitations, we could not include such useful information in the previous version. However, in the camera-ready version, we intend to include per-frame or per-sequence bounding-box distributions and distance bins and other meaningful visualizations to provide a clear overview of the dataset's content. Currently, the dataset’s diversity in motion, lighting, and occlusion is displayed in Table 3.
>
> 3 & 4- Thank you for your thoughtful suggestions. In section 5 of the revised version, we explicitly formalize the supported tasks, including multimodal monocular depth estimation and pedestrian detection with Event+RGB fusion tracking. In this new experiment, we evaluated the dataset on the two tasks mentioned and used different conditions such as lightning and the subsets' camera motions (static vs dynamic). The pedestrian detection is now conducted using both mAP50 and mAP50:95 metrics for static and dynamic subsets. Similarly, monocular depth estimation was evaluated across depth metrics under both subsets. These experiments help reveal the diversity and challenges of this dataset.
>
> We have addressed all your questions in the newest version of the article. We remain at your disposal for any additional information or suggestion.
>
> Thank you.

---

### Official Review · Reviewer_wXLm · 2025-10-30

**Soundness:** 2
**Presentation:** 2
**Contribution:** 2
**Rating:** 2
**Confidence:** 4

**Summary:**

The paper presents triCAM, a monocular multi-modal pedestrian dataset collected at two restaurant venues, combining event streams, RGB, depth, IMU, pedestrian boxes, and calibration. Sequences include static and dynamic camera motions. Events are binned at 33 ms to align with ~30 FPS depth frames, and the spatial/temporal synchronization pipeline is described in detail. The authors report YOLOv8x baselines for Event-only, RGB-only, and late fusion (NMS), with fusion outperforming single modalities in mAP metric. Despite the clear sync/calibration procedures and usable baselines, the paper has core issues: the motivation for the restaurant setting is weak, scale/diversity are limited, and there is no condition-wise analysis showing where events help (e.g., low light, fast motion, occlusion). The baselines and evaluation are insufficient (limited metrics/fusion strategies, no cross-scene generalization/transfer), and the overall presentation reads more like a careful engineering report than a dataset paper with compelling novelty. Therefore, I recommend rejection.

**Strengths:**

1. **Comprehensive multimodal setup with clear sync/calibration.** The platform jointly captures events, RGB, depth, and IMU, enabling research on cross-modal fusion and alignment. The paper details event-to-frame temporal alignment and spatial calibration procedures that are reasonably reproducible, lowering the barrier to use.

2. **Practical data organization.** The release includes pedestrian bounding boxes and starter training/evaluation scripts (e.g., YOLO), making it friendly for baseline reproduction and quick experimentation.

3. **Empirical indication of modality complementarity.** Although the analysis is not yet thorough, initial results show Event + RGB fusion outperforming single modalities, suggesting genuine potential for multimodal gains.

**Weaknesses:**

1. **Motivation for restaurant events is under-argued.** The paper cites event cameras’ advantages for high-speed and robust perception, but the collected restaurant scenes are not clearly high-speed. With only two venues, it is hard to claim representativeness for “socially interactive environments.” The paper does not concretely show which sub-conditions (e.g., strong backlight, low light, rapid motion, heavy occlusion, hand-held shake) in the dataset require events beyond what RGB can handle.

2. **Scale and diversity are limited.** The dataset contains ~20 sequences across two restaurants, while per-sequence stats are given, there is no explicit partitioning by motion speed or illumination, and no targeted “event-advantage” splits (e.g., low light, motion blur, fast actions).  Table 4 reports improvements on the entire test set only. Condition-wise statistics (e.g., low light, fast motion, dynamic camera, high occlusion) are necessary to determine where events are most beneficial, rather than relying on a single overall number.

3. **Label scope is narrow compared to stated goals.** Only bounding boxes for pedestrians are provided, tasks central to human-centered interaction (segmentation, pose/landmarks, ReID, trajectories/MOT) are absent, limiting the dataset’s relevance to the broader HRI/behavior understanding vision it outlines.

4. **Baselines and evaluation are thin.** Metrics stop at mAP@50, COCO mAP@[.5:.95] is not reported. Fusion is late-fusion NMS only, no early/mid-level or joint training fusion. No cross-scene generalization, and no evidence that models trained on triCAM can generalize better to other datasets/scenes.

5. **“Monocular” emphasis lacks compelling evidence.** The paper claims to be the first publicly available monocular multi-modal pedestrian dataset and contrasts with stereo-heavy rigs in related work, but it does not quantify the practical advantages of monocular over (i) using a single lens from a stereo rig or (ii) small-baseline stereo under the same setup (e.g., differences in calibration/sync complexity, cost, power, drift, failure cases).

**Questions:**

See weaknesses.

---

> ### Author Response · Authors · 2025-11-27
> **Clarifications on Data content and purpose**
>
> Thank you for your insightful and constructive comments. The revised version contains your constructive suggestions. We will address the highlighted weaknesses below:
>
> 1-**Motivation for restaurant events is under-argued**. We agree that the motivation for the restaurant might be judgmental. However, we find it very important to access a dataset that captures a natural interaction of our daily life where the motion is highly irregular, the occlusion and shakes are frequent, and the illumination varies with time. triCAM includes both static and dynamic camera motion where the camera is in fast motion.  The dynamic sequences were recorded with rapid handheld movements to capture fast human micro-motions and quick object interactions, conditions where event sensors naturally excel. As a multi-modal dataset, triCAM leverages the complementary strengths of RGB and event data: fast motion, low illumination, and camera shake often cause motion blur or noise in RGB frames, whereas the event camera captures these changes with high precision. These sub-conditions such as fast motion, low light, and handheld are explicitly described and motivated in the introduction of and Table 3 of the upated version. We kindly invite the reviewer to revisit these sections for full context.
>
> 2-**Scale and diversity are limited**. Table 3 contains all additional information about the content of this dataset. Maybe our explanation is not clear enough, but each sequence was recorded at a different time and environment. Most indoor recordings are captured in low light.
> Regarding the lack of robust experimentation, we conducted additional experiments on the two subsets (static and dynamic) of this dataset. The first one is a pedestrian detection algorithm evaluated using two metrics: the mPA50 and mAP50:95. The second one is monocular depth estimation using HMnet.  These new experiments are discussed with detailed information about the model training procedure in section 5 of the revised version of the article.
>
> 3- **Label scope is narrow compared to stated goal**. We agree that the original description of the dataset’s broader goals was overly ambitious. In practice, heavy occlusions, rapid body motion, and high density made pose, ReID, and trajectory annotations unreliable, even with semi-automated tools. For this reason, we removed these label types and focused on delivering high-quality pedestrian bounding boxes, which remain the most stable and widely used annotation for crowded social environments.
> The revised article now makes this scope explicit and positions TriCAM as a pedestrian-centric detection dataset, not an all-purpose HRI benchmark.
>
> 4- **Baseline and evaluation are thin**. We expanded the experimental section substantially. Section 5 of the revised version now includes:COCO mAP50:95 on pedestrian detection and a new monocular depth estimation on both dynamic and static subsets as explained in the response to W2. Additional information strategies such as low-light, fast-motion and occlusion  are presented in Table 3.
>
> 5- **Monocular emphasis lacks compelling evidence**. These points were already stated in the Introduction, where we highlight the well-known advantages of event cameras including fast motion, low light, and robustness to camera shakes which explain their recent surge in popularity.  We also clarify the motivation for the monocular setup, as it significantly reduces calibration and synchronization complexity, as well as power, size, and cost, compared to stereo rigs. We briefly explain the comparision between monocular and small-baseline stereo configurations and explain why stereo brings limited benefits in close-range or crowded scenes in the Introduction. Finally, note that TriCAM is the first publicly available monocular RGB-Event pedestrian dataset, which justifies this design choice. Our objective is to offer a multimodal dataset that facilitate the development of lightweight applications.
>
> Thank you.

---

### Official Review · Reviewer_ouX8 · 2025-10-31

**Soundness:** 2
**Presentation:** 2
**Contribution:** 2
**Rating:** 2
**Confidence:** 4

**Summary:**

This paper proposes a monocular multi-modal pedestrian dataset, triCAM. The dataset is captured with RGB, depth, event, and IMU sensors in two different restaurants. The images are further manually annotated by humans. In the experimental results, the mAP
performance using YOLOv8 is provided. The hardware, software setup, and calibration process are also introduced.

**Strengths:**

- The collected triCAM provides a new multi-modality benchmark to the pedestrian detection community.
- The collecting and post-processing pipeline is introduced clearly.

**Weaknesses:**

- The major concern is about privacy protection. As a pedestrian dataset, it is not mentioned whether permission was obtained from the subjects, or whether the dataset will be made public, or under what license.
- The dataset used manual annotations. But the annotation compensation was not mentioned, so I also added a corresponding flag for the ethics review.
- This dataset is limited to two specific restaurant scenarios. Therefore, its general applicability is limited. The existing datasets mentioned in Table 1 cover more general situations than this dataset.
- The paper only reported the performance on a specific object detector, which fails to reflect the challenge of this new dataset and the necessity of releasing such a dataset.

**Questions:**

Please refer to the weaknesses.

**Details Of Ethics Concerns:**

As mentioned, this paper did not mention whether permission was obtained from the subjects, or whether the dataset will be made public, or under what license. Meanwhile, the dataset used manual annotations. But the annotation compensation was not mentioned.

---

> ### Author Response · Authors · 2025-11-27
> **Clarification on privacy protection**
>
> Thank you for pointing out these important concerns. We apologize for not describing the privacy and ethics procedures clearly in the previous version.  We will address the highlighted weakness below:
>
> - Regarding the privacy and consent, we reassure you that all the participants of the TriCAM data collection signed a consent form approved under our institution’s GDPR compliant ethics protocol. No personal information is stored or released. All identifiable facial features were automatically anonymized using a face detection pipeline followed by Gaussian blurring to ensure that no identities can be recognized, as stated from line 375 to 382 in the revised version.
>
> - We apologize for not clarifiing this points. All manual annotations were carried out directly by us, the authors of the paper. Therefore, no external annotators were hired as tated in Section 4.1.
>
> - We acknowledge that triCAM covers two restaurant environments. This dataset is therefore intentionally designed with a focus scope rather than a general-purpose data. The manuscript has been updated to clearly state this limitation and to position triCAM as a specialized benchmark for socially interactive indoor and outdoor scenarios in the Introduction.
>
> - Thank you for your thoughtful remark, we have expanded our experimental section in the revised article to include additional detectors and evaluation metrics, demonstrating more clearly the challenges posed by triCAM and supporting the relevance of releasing this dataset in Section 5.
>
> These clarifications have been added to the newest article version to address the reviewer’s concerns regarding ethical compliance, privacy, dataset scope, and baseline evaluation.
>
> Thank you.

---

### Meta-Review · Area_Chair_hf8w · 2025-12-24

**Summary:**

This paper introduces the triCAM dataset, a monocular multimodal pedestrian dataset collected in a restaurant environment, containing synchronized RGB, depth, event, and IMU data, as well as manually annotated pedestrian bounding boxes. The authors designed the hardware platform, calibration and synchronization procedures, and data format, and provided baseline pedestrian detection results using YOLOv8 in both single-modality and simple late-fusion settings.

The advantages of this paper are as follows:
1. This paper introduces a new multimodal dataset which is relatively rare in the fields of pedestrian detection and event-based vision.
2. Preliminary baseline experiments suggest that event data and RGB images may be complementary in pedestrian detection.

The disadvantages pointed out by the reviewers are as follows:
1. The motivation of this work is unclear. Why event cameras are needed for the pedestrian detection task?
2. The experimental evaluation is weak, using only a single detector, limited evaluation metrics, and a simple post-fusion method, lacking analysis under different conditions or multi-task benchmarking.
3. Several reviewers raised serious ethical and privacy concerns, including the lack of discussion on subject consent, data release policies, licensing agreements, and compensation for annotators.
4. Some related work are missing. There are also some event-based camera datasets and algorithms related to human body capture.

[1] From Sharp to Blur: Unsupervised Domain Adaptation for 2D Human Pose Estimation Under Extreme Motion Blur Using Event Cameras

[2] A temporal densely connected recurrent network for event-based human pose estimation

[3] EventEgo3D: 3D Human Motion Capture from Egocentric Event Streams

[4] Reli11d: A comprehensive multimodal human motion dataset and method

[5] EventHPE: Event-based 3D Human Pose and Shape Estimation

[6] Efficient Human Pose Estimation via 3D Event Point Cloud

**Reviewer Concerns:**

This dataset paper lacks sufficient innovation and a compelling motivation. It reads more like an engineering report than a noteworthy benchmark contribution.

The reviewers are concerned with privacy and responsible research practices.

**Reviewer Scores:**

Most reviewers recommended "reject" or "marginally reject," with only one reviewer giving a slightly positive evaluation. The reviewers are concerns with the the limited contribution and insufficient evaluation issues. I think the reviewers will not change their opi

---

### Decision · Program_Chairs · 2026-01-26

Reject